# In-vivo efficacy of biodegradable ultrahigh ductility Mg-Li-Zn alloy tracheal stents for pediatric airway obstruction

Jingyao Wu[1], Leila J. Mady[2], Abhijit Roy [1], Ali Mübin Aral[3], Boeun Lee[1], Feng Zheng[4], Toma Catalin[5,6], Youngjae Chun [1,6,7], William R. Wagner[1,3,6,8], Ke Yang[4], Humberto E. Trejo Bittar[9], David Chi[10] & Prashant N. Kumta[1,6,8,11,12,13 ✉]

Pediatric laryngotracheal stenosis is a complex congenital or acquired airway injury that may manifest into a potentially life-threatening airway emergency condition. Depending on the severity of obstruction, treatment often requires a combination of endoscopic techniques, open surgical repair, intraluminal stenting, or tracheostomy. A balloon expandable biodegradable airway stent maintaining patency while safely degrading over time may address the complications and morbidity issues of existing treatments providing a less invasive and more effective management technique. Previous studies have focused on implementation of degradable polymeric scaffolds associated with potentially life-threatening pitfalls. The feasibility of an ultra-high ductility magnesium-alloy based biodegradable airway stents was demonstrated for the first time. The stents were highly corrosion resistant under in vitro flow environments, while safely degrading in vivo without affecting growth of the rabbit airway. The metallic matrix and degradation products were well tolerated by the airway tissue without exhibiting any noticeable local or systemic toxicity.

[1] Department of Bioengineering, University of Pittsburgh, Pittsburgh, PA 15261, USA. [2] Department of Otolaryngology, University of Pittsburgh, Pittsburgh, PA 15261, USA. [3] Department of Surgery, University of Pittsburgh, Pittsburgh, PA 15261, USA. [4] Institute of Metal Research, Chinese Academic of Sciences, Shenyang 110016, China. [5] Heart and Vascular Institute, University of Pittsburgh Medical Center, Pittsburgh, PA 15261, USA. [6] McGowan Institute of Regenerative Medicine, Pittsburgh, PA 15261, USA. [7] Department of Industrial Engineering, University of Pittsburgh, Pittsburgh, PA 15261, USA. [8] Department of Chemical and Petroleum Engineering, University of Pittsburgh, Pittsburgh, PA 15261, USA. [9] Department of Pathology, University of Pittsburgh Medical Center, Pittsburgh, PA 15213, USA. [10] Department of Otolaryngology, Children's Hospital of Pittsburgh of UPMC, Pittsburgh, PA 15224, USA. [11] Department of Mechanical Engineering and Materials Science, University of Pittsburgh, Pittsburgh, PA 15261, USA. [12] Department of Oral Biology, School of Dental Medicine, University of Pittsburgh, Pittsburgh, PA 15261, USA. [13] Center for Complex Engineering Multi-functional Materials, University of Pittsburgh, Pittsburgh, PA 15261, USA. ✉email: pkumta@pitt.edu

Laryngotracheal stenosis (LTS) and subglottic stenosis (SGS) are challenging adult and pediatric conditions[1,2] requiring endoscopic, tracheostomy tube placement, and/or open surgical repair[1,3]. Severely stenotic pediatric patients undergo multiple temporizing endoscopic procedures with ~68% requiring open surgical revision[2] typically accompanied by recurrent scarring, poor wound healing, or life-threatening anastomotic disruption[4] with temporary intraluminal airway stent commonly providing graft stability and reinforcement[5]. Pediatric stenting is an unproven primary treatment option with commercial metal or silicone derived tracheal stents also incapable of optimal clinical outcomes, even in adults. Advanced biomaterials based airway stents could offer a less invasive, reduced morbidity successful treatment option.

Preclinical and clinical studies report using degradable biocompatible polymeric tracheal stents[6–22] of poly(lactic-co-glycolic acid) (PLGA), poly(L-lactic acid) (PLLA), polycaprolactone (PCL)[19] with polydioxanone ubiquitously used[8–10,23] for acute adult and pediatric airway obstruction treatments[8,9,22,24]. However, stent migration, repeated stenting, degradation fragment retention, and fatal dyspnea episode events are reported complications. Nevertheless, 3D printed external tracheal splints alleviate obstructive symptoms without inhibiting primary airway growth[25–28]. Current biodegradable tracheal stents mimicking commercial non-degradable metallic stent design are ineffective for pediatric airway obstruction opening the doors for biodegradable metals for LTS and SGS treatment. Accordingly, pediatric tracheal stents of biodegradable magnesium alloy was recently demonstrated in rabbit trachea[29]. To date however, there are no reports on high ductility biodegradable magnesium alloy for tracheal stent applications.

Herein, for the first time, we demonstrate the superior response of a balloon-expandable ultra-high ductility (UHD) biodegradable magnesium tracheal stent contrasted with traditional non-degradable stents. We reported implantation success of prototype magnesium alloy stent in a rat tracheal bypass model[30] and Mg-Li-Zn multiphase UHD alloys for stent application[31] with Mg-6Li-1Zn (LZ61) demonstrating greatest clinical application potential owing to its superior mechanical properties, low degradation rate, UHD of ~50% and excellent biocompatibility. Hence, the LZ61-KBMS (Kumta Bioresorbable Magnesium Stents) alloy series of tracheal stents were fabricated and tested both in vivo and in vitro. The corrosion resistance of LZ61-KBMS was evaluated in a bioreactor simulated dynamic flow environment juxtaposed with AZ31 control stents, a commercial high corrosion-resistant magnesium alloy. LZ61-KBMS were further implanted into a healthy rabbit trachea for in vivo degradation and airway tissue response evaluation. 316L stainless steel (316L SS), commonly used commercial non-degradable stents[32–35], served as the control. Our successful results provide early stage preclinical evidence validating the LZ61-KBMS stents potential for pediatric LTS treatment.

## Results
**Tracheal stent fabrication.** As-cast LZ61-KBMS was extruded (Fig. 1) into 20 mm diameter microstructurally refined rod (Fig. 1A (i, ii)) followed by mini tube fabrication (Fig. 1B) using wire-electrical discharge machining (wire-EDM)[36–38]. The LZ61-KBMS alloy mini tubes (Fig. 1B (i, ii) (4.2-mm diameter, 60-mm length, and 300-μm wall thickness) cut from rods (Fig. 1B (iii, iv) were laser cut into LZ61-KBMS following a programmed pattern (Fig. 1C (i, ii, iii)) giving granular laser-cut surfaces (Fig. 1C (iv, v) contrasted to relatively smoother wire-EDM cut surfaces. All stents were hence, electrochemically polished (Fig. 1D (i)) hooking LZ61-KBMS to a pure Mg wire, connected to a DC anode reducing inhomogeneous polishing at LZ61-KBMS contact spots and producing polished

LZ61 KBMS with smooth metallic lustre stent strut of reduced surface granularity (Fig. 1D (ii, iii, iv)). AZ31 and 316L SS stents were also similarly fabricated, and the corresponding stent surface images are shown in the Supplementary section, Supplementary Fig. 1.

**In vitro degradation of LZ61 stent in a bioreactor.** We earlier published the degradation of our novel LZ61 alloy matching AZ31 under static environments[31]. However, in vivo, tracheal stents are exposed to a dynamic fluid environment with moving mucus consistently cleaning the airway. Therefore, it is necessary to examine LZ61-KBMS degradation compared to commercial AZ31 stents under dynamic flow condition before in vivo implantation. Accordingly, LZ61-KBMS and AZ31 stents were delivered into silicone tubes and immersed in Hanks' balanced salt solution (HBSS) with the dynamic flow for 1, 3 or 5 weeks. 3D reconstruction of μCT stent scans at different time points (Fig. 2A) shows that LZ61-KBMS degraded significantly slower than the control AZ31 stents, under dynamic HBSS flow. The LZ61-KBMS structure was maintained even after up to five weeks of immersion in dynamic flow, whereas the AZ31 stents were almost completely degraded. In addition, calculation of the stent volume remaining (Fig. 2B) indicates 47.6% of the LZ61-KBMS degraded while only 7.8% of the total volume was left for AZ31 stents. Wang et al. reported the contribution of flow-induced shear stress to increased degradation of Mg alloys due to oxidation layer breakage and prevalence of more localized corrosion[39]. Grogan et al.[40] also reported that flow conditions increase AZ31 alloy corrosion rates relative to static tests. Our results demonstrate that LZ61-KBMS alloy is more corrosion resistant under condition of flow-induced shear stresses. After 5 weeks of immersion, SEM (Fig. 2C (i, ii)) show stent surface covered by degradation layer and precipitates from HBSS, especially at major degradation locations. EDAX (Fig. 2D) results indicate higher Ca and P deposits on LZ61 stents surface implying a more stable Ca and P degradation layer formation compared to AZ31, known to protect the underneath alloy matrix and thus improving the corrosion resistance[41,42].

**In vivo degradation of LZ61-KBMS stent in rabbit tracheal model.** Efficacy and bio-toxicity of the LZ61-KBMS tracheal stents were tested in normal New Zealand rabbit trachea model chosen based on literature report[43]. The LZ61-KBMS or 316L SS stent (Fig. 3A) was placed beneath the first airway cartilage ring confirmed by X-ray imaging of the rabbit neck (Fig. 3B). LZ61-KBMS is hardly visible under the X-ray due to the lower density of the LZ61-KBMS alloy (Fig. 3B (ii)). Endoscopic (Fig. 3C) and optical coherence tomography (OCT) imaging (Fig. 3D) were performed at the end of each time point with endoscopic imaging of LZ61-KBMS at week 4 for monitoring stent structure and position.

The OCT images indicated LZ61-KBMS stent encapsulation in the airway tissues with visible gas pockets due to $H_2$ gas release during stent degradation contributing to the airway tissues porosity surrounding the stents. The gas pocket formation is common for magnesium alloy-based implants[44–46], but their effect on airway tissue is unknown to date. The airway stented with 316L SS stent appeared similar to LZ61-KBMS under the endoscope except as expected, there was no gas pocket around the 316L SS stent validated by the OCT images. At 8 weeks, for the LZ61-KBMS, there was no visible stenotic tissue formation, and the trachea lumen appeared healthy. OCT scanning also further verified complete stent degradation with airway lumen restoration to its normal smooth surface. In contrast, endoscopy imaging indicated steady stenotic tissue growth in the 316L SS

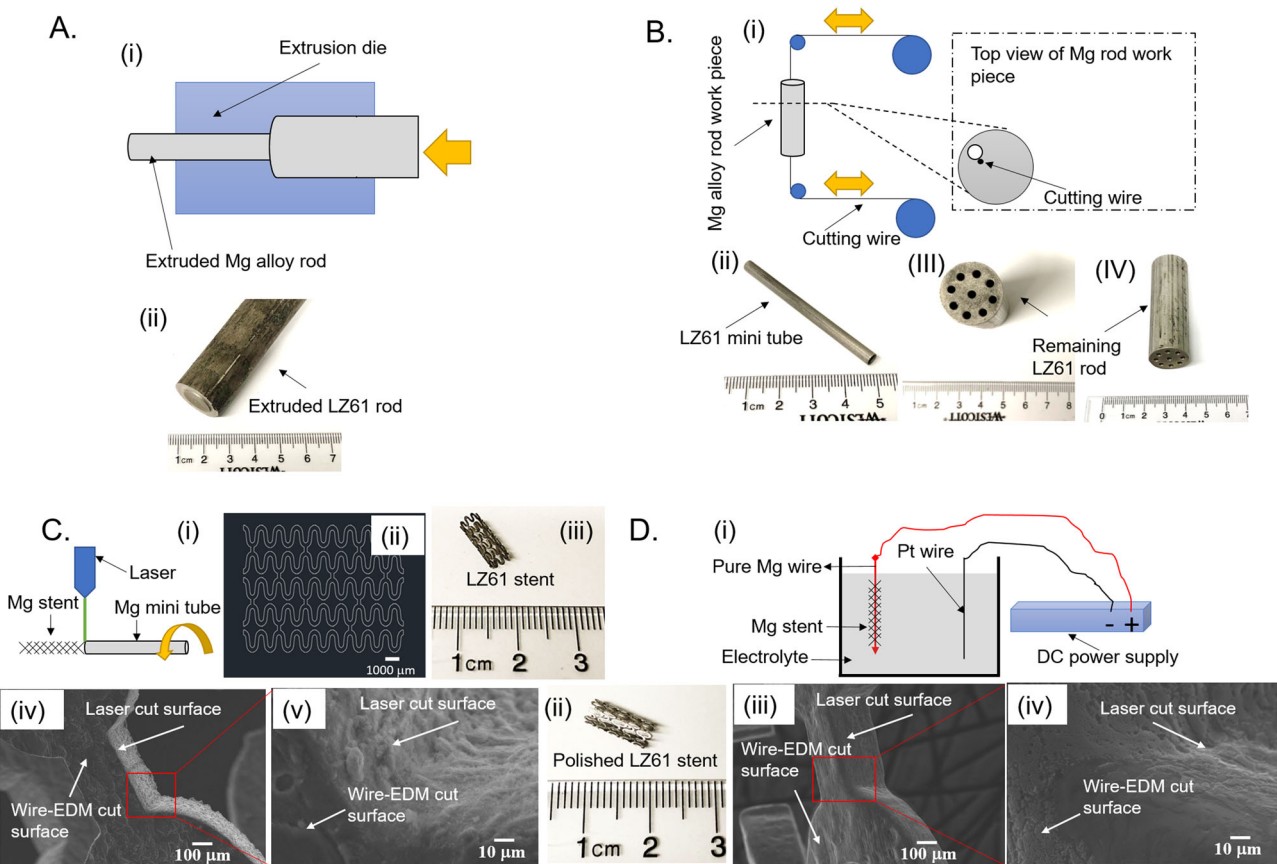

**Fig. 1 Schematic illustration of the stent fabrication process. A** (i) LZ61-KBMS alloy extrusion process, (ii) photograph of an extruded LZ61-KBMS rod.
**B** (i) Fabrication of LZ61-KBMS mini tubes using wire-electrical discharge machining (wire-EDM). A hole was drilled all the way through the extruded Mg rod, and then the wire was inserted through the hole, and cut the work piece when a high voltage was applied. (ii) photograph of a LZ61-KBMS mini tube (4.2 mm in diameter, 60 mm in length and 0.300 mm in wall thickness) cut by wire EDM, (iii, iv) remaining part of the extruded LZ61-KBMS rod after wire-EDM, the empty holes are the spots where the mini tube was cut off from. **C** (i) Laser cutting process. The LZ61-KBMS/AZ31/316L SS mini tube was fixed by the fixture and a laser beam was exposed vertical to the mini tube surface and cut through the tube wall. The fixture moved the tube following the pattern designed for the stent as the laser was cutting the mini tube. (ii) The stent design and 2D pattern the laser beam followed while cutting the mini tube. (iii) Photograph of a laser-cut LZ61-KBMS stent. (iv) SEM image of the LZ61-KBMS stent strut after laser cutting. (v) Higher magnification SEM image of the LZ61-KBMS stent strut after laser cutting. Rough surface could be observed on the surface cut by the laser or wire-EDM. **D** (i) Electrochemical polishing of the stent. The LZ61-KBMS/AZ31 stent was hooked on a pure Mg wire that was connected to the anode. Pt wire served as the cathode. Both the stent and Pt wire were immersed in the acid electrolyte and the electrochemical polishing was carried out using a fixed DC current. (ii) Photograph of the LZ61-KBMS stent after electrochemical polishing. The stent surface clearly displayed the metallic sheen after polishing. (iii) SEM image of the LZ61-KBMS stent strut after electrochemical polishing. (iv) Higher magnification SEM image of the LZ61-KBMS stent strut after electrochemical polishing.

stent group at both weeks 8 and 12 in line with the 8 and 12 weeks OCT results. Stenotic tissue growth is typical for non-degradable metallic stents[47]. It is hence, critical to understand the stent's implantation impact on tracheal lumen size in growing rabbits, since this is a key indicator of normal pediatric patient respiratory function after receipt of tracheal stent implants over the stent degradation period and its effect and interference on overall natural tracheal growth. Therefore, we calculated the tracheal lumen size with the stents implanted inside based on the OCT images using the Image J software (Fig. 3E).

The lumen size of the LZ61-KBMS implanted trachea increased steadily with time as the rabbits continued to grow confirming no adverse impact on the growth upon LZ61-KBMS implantation and degradation. In contrast, with the 316L SS stents, the airway lumen size decreased with continued whole airway growth (Fig. 3E). The airway lumen size of the 316L SS control was significantly smaller than the LZ61-KBMS group at weeks 8 and 12 ($p < 0.01$) due to granulation tissue growth observed by endoscopy and OCT imaging of the non-degradable control group. The 3D structures of the remaining stents reconstructed

based on the µCT scan (Fig. 3F) shows expectedly stable 316L SS stent structure preservation over the entire implant duration. However, for the LZ61-KBMS group, $53 \pm 12\%$ of the initial volume degraded after 4 weeks, fully degrading at week 8 supporting the endoscopy and OCT imaging results.

**Histology analysis.** Figure 4A–C shows the H&E staining of airway cross-section for the LZ61-KBMS, 316L SS and the healthy rabbit control groups. At week 4, the LZ61-KBMS stents were embedded in the airway tissue with portions covered by epithelium (Fig. 4A (i, ii)). Dark hematoxylin stain surrounding the LZ61-KBMS stent strut showed a higher cell density indicating potential inflammatory response at early implantation stages. The tracheal cross section was largely circular due to the radial force from the expanded LZ61-KBMS stent (Fig. 4A (i)) at week 4. The epithelium layer at weeks 8 (Fig. 4A (iv)) and 12 (Fig. 4A (vi)) was fully restored representative of the normal healthy rabbit airway tissue (Fig. 4C (i)). The tracheal cross-sectional shape was elliptical at weeks 8 (Fig. 4A (iii)) and 12 (Fig. 4A (v)) representative of normal rabbit trachea shape (Fig. 4C (i)). Large vessels are seen

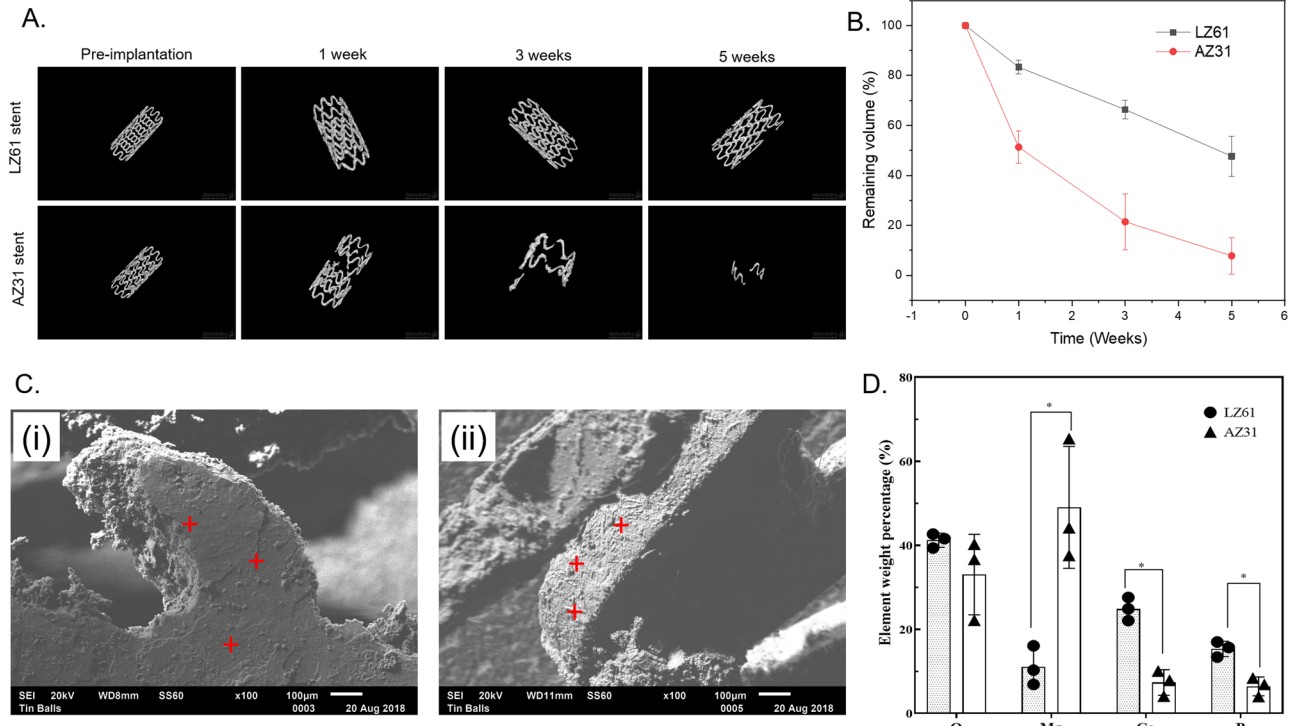

**Fig. 2 In vitro degradation of LZ61-KBMS stents in bioreactor. A** The 3D structure of AZ31 and LZ61-KBMS stents based on µCT scan at different time points indicating the remaining stents after immersion in a dynamic flow environment. **B** the remaining volume of the stents calculated based on the 3D structure of AZ31 and LZ61-KBMS stents. (n = 3). **C** The SEM images of the stent surface after 5 weeks of immersion in a dynamic flow environment (i) LZ61-KBMS stent, (ii) AZ31 stent, scale bar = 100 µm. **D** The presence and elemental percentages of O, Mg, Ca, and P present in the degradation layer deposited on the stents after 5 weeks of immersion as measured by EDAX. (n = 3, *P < 0.05).

in the submucosa at weeks 8 (Fig. 4A (iv)) and 12 (Fig. 4A (vi)). The 316L SS stent hardness made it difficult to cut the paraffin block without tearing the tissue around the stent. Therefore, all 316L SS stents were removed from the trachea before commencing paraffin embedding. At week 4, similar to LZ61-KBMS, the tracheal cross-session was circular due to the radial force from the expanded 316L SS stent (Fig. 4B (i)). As the rabbit trachea grew larger commensurate with the rabbit age in time, the trachea lumen also restored to the normal elliptical shape (Fig. 4B (iii, v)). A healthy intact epithelium layer was observed at all-time points (Fig. 4B (ii, iv)) except for the histology collected at week 12 (Fig. 4B (vi)). Part of the epithelium layer was lost due to the removal of the 316L SS stent. H&E stain examinations of the key organs (lung (Supplementary Fig. 2), kidney (Supplementary Fig. 3) and liver (Supplementary Fig. 4) evaluated systemic effects related to stent degradation. Stained sections did not demonstrate any pathology or toxicity evidence. Further, no differences were observed when compared to the corresponding healthy rabbit organs (Supplementary Figs. 2–4).

Mucus and goblet cells presence was clearly identified using Alcian blue staining (Fig. 5). For LZ61-KBMS, a thick mucus layer (seen in blue color) was observed at week 4 (Fig. 5A (i)), likely indicative of more mucus secretion due to the existence of LZ61-KBMS. In the submucosa region where the stent strut existed, there was also a higher blue stain density, reflective of a non-specific staining indicative of possible fibrotic tissue formation around the LZ61-KBMS stents. However, at weeks 8 (Fig. 5A (ii)) and 12 (Fig. 5A (iii)), normal epithelium and mucus secretion was observed for the LZ61-KBMS stent group when compared to the normal trachea (Fig. 5C). Removal of the 316L SS stent made it impossible to identify the precise stent strut location. However, the epithelium and mucus secretion appeared

normal (stained image in Fig. 5B (i–iii)). Goblet cells were evenly distributed in all groups at all-time points similar to the normal trachea (Fig. 5C). Goblet cell quantification along the epithelium in randomly selected regions plotted (Fig. 5D shows no statistically detectable difference among all the groups.

Histiocytes presence was further analyzed with CD68 immunostaining (Fig. 6). Histiocytes clusters (red circle marked region) were identified in LZ61-KBMS group at week 4 (Fig. 6A (i)) and at all-time points in the 316L SS group (Fig. 6B (i, ii, iii)). Full degradation of LZ61-KBMS resulted in no histiocytes clusters and the stained sections were similar to the healthy rabbit control tracheal tissue (Fig. 6C).

## Discussion

Currently available commercial tracheal stents fail to deliver long term satisfactory clinical outcomes. The objective here was to evaluate if the novel magnesium-based UHD biodegradable stents are superior to current non-degradable metallic stents. The biomechanical performance and ability of the stent to maintain airway patency over time was well controlled by the LZ61-KBMS alloy corrosion in turn controlled by the alloy chemical composition, the surrounding electrolyte, and the prevalence of external stress. We have reported the impact of alloy chemical composition and surrounding electrolyte on Mg-Li-Zn alloy corrosion[31]. However, the impact of the flow-induced shear stress and internal stress on corrosion of the LZ61-KBMS stent was not studied. Hence, the fabricated LZ61-KBMS stents were tested in a bioreactor as well as in a healthy rabbit tracheal model.

In the bioreactor study, LZ61-KBMS demonstrated much better corrosion resistance when compared to commercial AZ31 (Fig. 2). Wang et al. proved that flow-induced shear stress plays

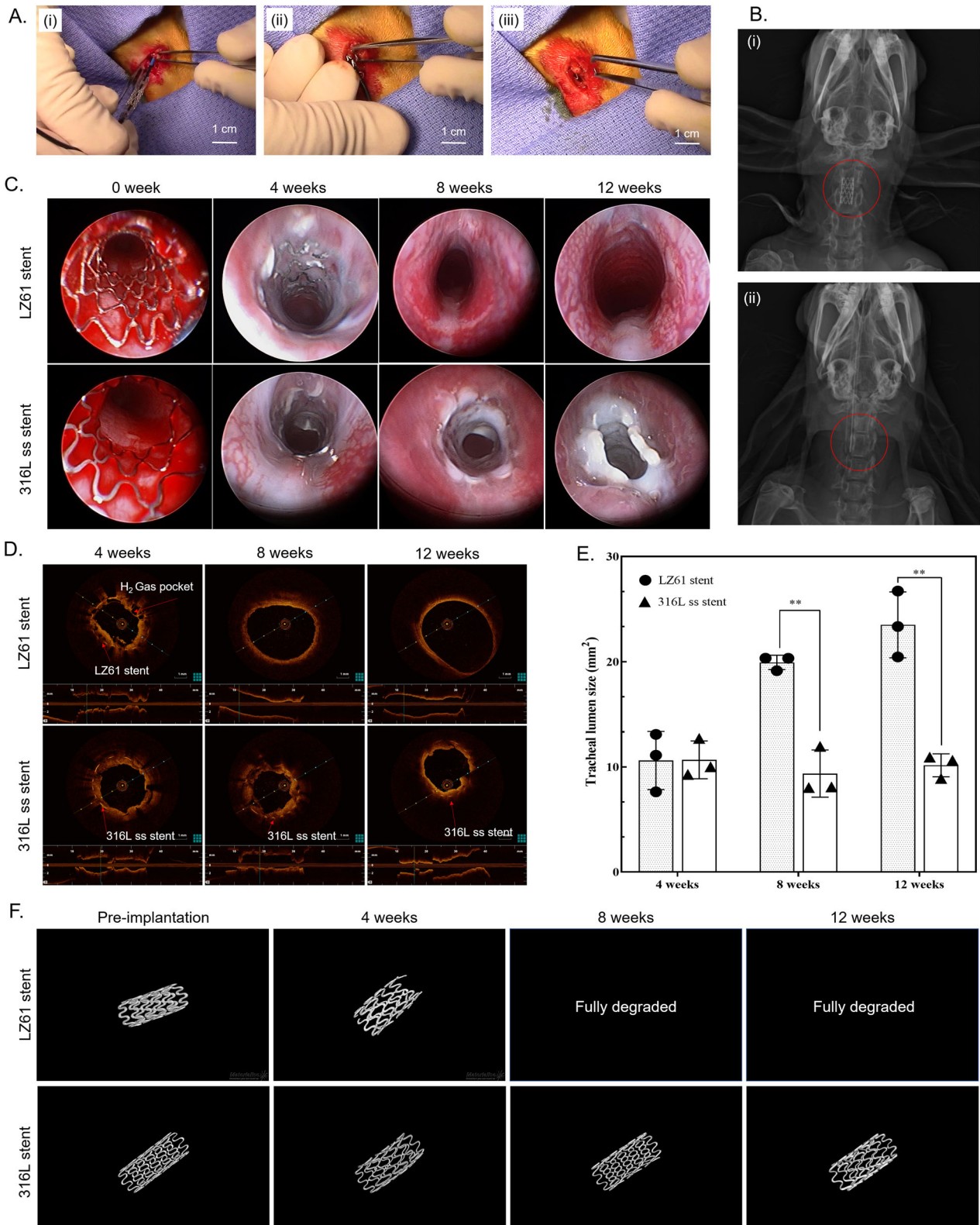

**Fig. 3 In vivo implantation of LZ61-KBMS stents in a healthy rabbit tracheal model. A** The surgical procedure of the stent placement. (i) Tracheal stent was mounted on the balloon. (ii) The balloon was injected with saline and expanded the tracheal stent against the tracheal wall. (iii) The balloon was deflated and pulled out of the trachea leaving the stent inside the tracheal lumen**. B** X-ray image of the implantation site of the (i) 316L SS stent and (ii) LZ61-KBMS stent after the tracheal stent implantation. The positions of the stents are circled in the images. LZ61-KBMS stent is hard to identify here under the X-ray due to the lower density. A dark shadow is only visible here. **C** Endoscopic images of the stented airway right after implantation, and after 4, 8, and 12 weeks of implantation. **D** OCT images of the cross-section of the stented airway at 4, 8, and 12 weeks after implantation. **E** Tracheal lumen area calculated based on the cross-section of OCT images at 4, 8, and 12 weeks. **denotes a significant difference between alloy groups ($p < 0.01$, $n = 3$). **F** The 3D structure of 316L SS stents and LZ61-KBMS stents based on µCT scan at different time point indicating the remaining stents after implantation in vivo.

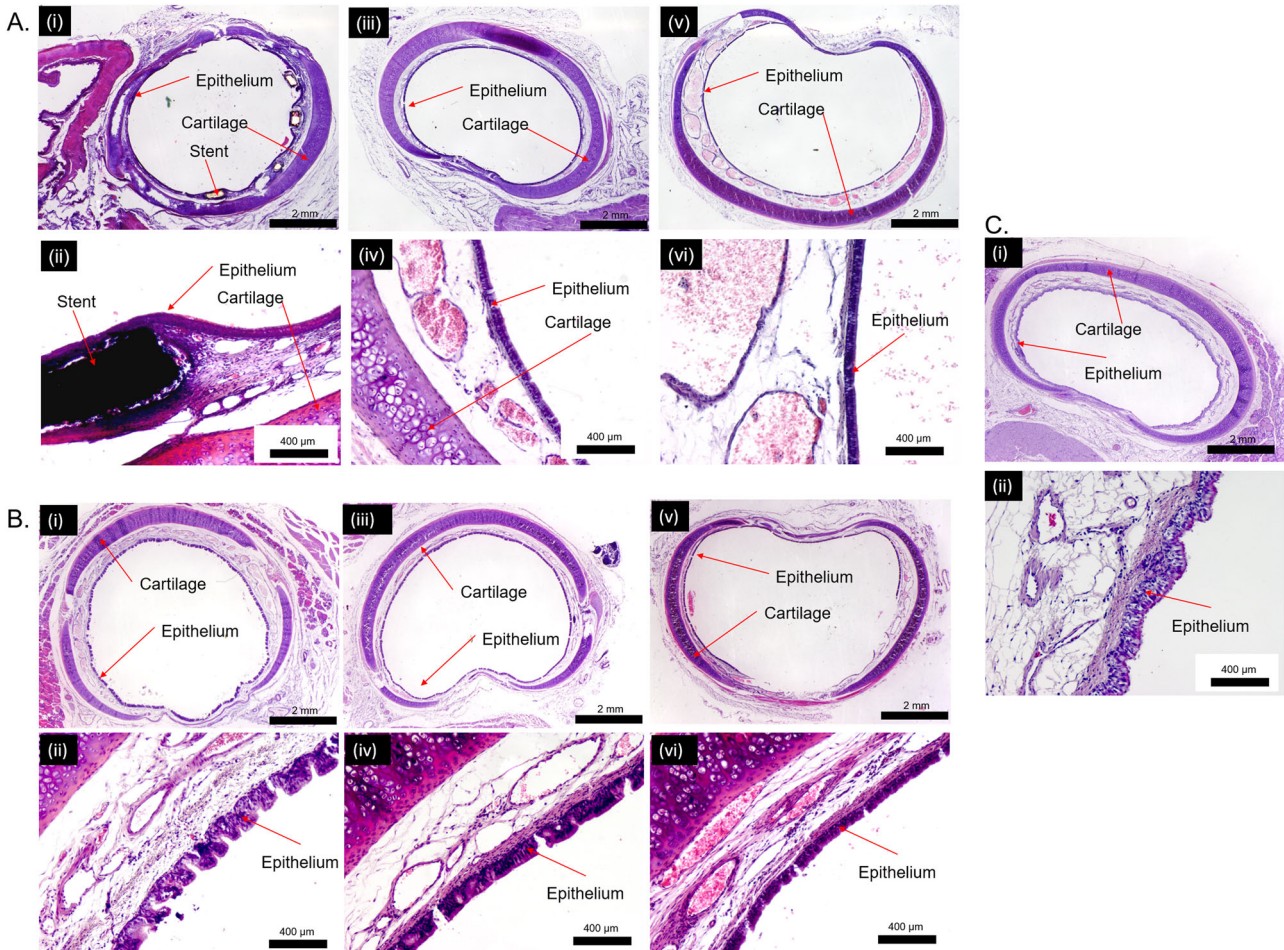

**Fig. 4 H&E staining of the stented rabbit trachea. A** LZ61-KBMS stent stented trachea at various magnifications after (i–ii) 4 weeks, (iii–iv) 8 weeks; (v–vi) 12 weeks, **B** 316L ss stent stented trachea at various magnifications after (i–ii) 4 weeks, (iii–iv) 8 weeks; (v–vi) 12 weeks, **C** (i–ii) healthy rabbit trachea at two different magnifications.

an important role in Mg alloy degradation[39]. Grogan et al.[40] also reported that flow conditions significantly increased AZ31 alloy corrosion rate relative to the conventional static tests. Following proposed three mechanisms explain the corrosion rates increase due to flow-induced shear stress: (1) Prevention of uniform corrosion layer formation; (2) Causation of more localized corrosion; and (3) Detachment of corrosion layer. For the LZ61-KBMS, more Ca and P was deposited into the degradation layers, conceivably more adherent and stable than the degrading AZ31 stent layers. Therefore, LZ61-KBMS were better protected under the flow environment, displaying a much slower corrosion rate.

Figure 7 summarizes the corrosion rates of LZ61-KBMS alloy and AZ31 alloy under different tests based on the current study and our previous publications[31]. Since AZ31 stents were not implanted in vivo herein, no results of AZ31 in vivo stent degradation rates are plotted in Fig. 7. However, fabrication of the LZ61-KBMS and AZ31 alloy stents led to significantly increased degradation rates compared to the corrosion rates measured from previous in vitro and in mice sub-cutaneous studies. This is attributed to three reasons: first, the surface area to weight ratio is much larger for stents offering a larger metal matrix area exposure to the corrosive environment; second, the electrochemical polished surface is different from the sand paper polished surface of the disks. Finally, there were no further heat treatments or microstructure refinement conducted on the as-

received stents following wire-EDM and laser cutting. These processes could induce significant strain hardening and the laser cutting process could easily induce varying heat affected zone states likely altering the bulk and surface microstructure yielding coarse, fine, and mixed grains affecting corrosion. Some of the surface structure modifications is likely removed during electrochemical polishing. The altered bulk microstructure is however preserved, which can still affect the overall corrosion rate. Further, there was no difference in the average corrosion rate of the LZ61-KBMS in the bioreactor after 5 weeks and that determined from the stent implanted in the rabbit trachea model after 4 weeks suggesting the bioreactor environment likely mimicking the critical parameters encountered in vivo. These results also indicate that the bioreactor conditions reflected the dynamic fluid flow across the stent as opposed to static conditions prevalent in the in vitro tests of the same alloys tested in the disk form. At the same time, in our publication[31], when both LZ61-KBMS and AZ31 alloys were tested as disks implanted in the mouse sub-cutaneous model, the corrosion rates exhibited no statistical difference. Hence, it appears that the bioreactor environment is more representative of the actual in vivo tracheal conditions.

The current study demonstrates the complete degradation of LZ61-KBMS without impacting further airway growth. At the same time, there is a steep increase in the degradation after the initial stage (~4 weeks) considering 48 ± 8% of the LZ61-KBMS

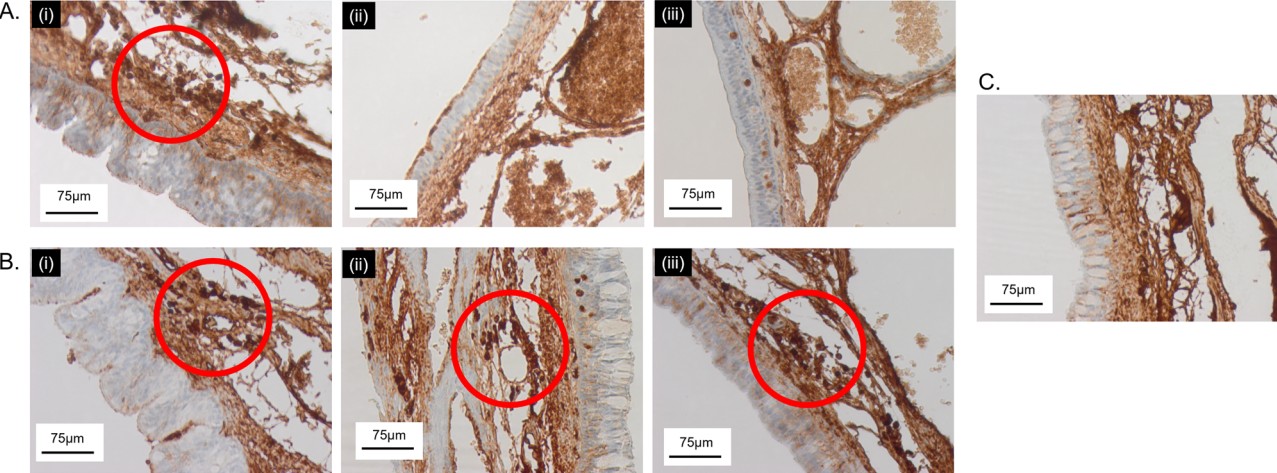

**Fig. 5 Alcian blue staining of stented rabbit trachea. A** LZ61-KBMS stent stented trachea. (i) 4 weeks, (ii) 8 weeks and (iii) 12-weeks post implantation, **B** 316L SS stent stented trachea (i) 4 weeks, (ii) 8 weeks and (iii) 12-weeks post implantation, **C** Healthy trachea, **D** number of goblet cells per 1 mm along the epithelium. No statistical difference was identified among the different groups ($P > 0.05$).

**Fig. 6 CD68 staining of stented rabbit trachea. A** LZ61-KBMS stent stented trachea. (i) 4 weeks, (ii) 8 weeks, and (iii) 12-weeks post implantation, **B** 316L SS stent stented trachea. (i) 4 weeks, (ii) 8 weeks, and (iii) 12-weeks post implantation, **C** healthy rabbit trachea. Red circled areas are spots where clusters of macrophage cells were detected.

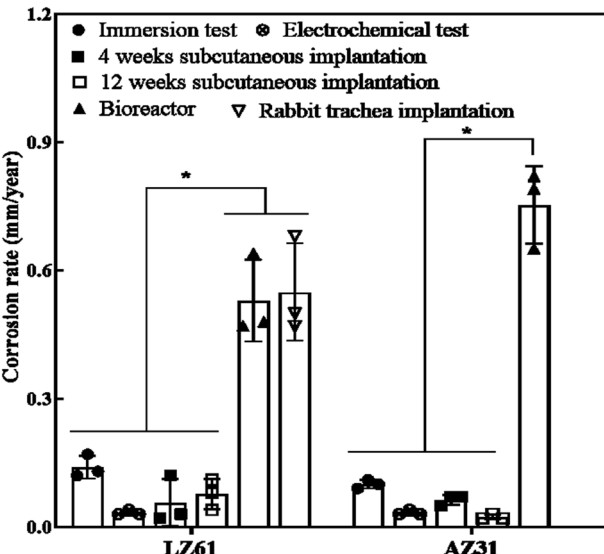

**Fig. 7 A summary of corrosion rate of LZ61-KBMS and AZ31 alloy under different tests.** Immersion test: 5-week corrosion rates calculated based on the immersion test; Electrochemical test: corrosion rates calculated based on Tafel plot; 4-weeks subcutaneous implantation: corrosion rates calculated based on the weight loss after 4 weeks of implantation; 12-weeks subcutaneous implantation: corrosion rates calculated based on the weight loss after 12 weeks of implantation; Bioreactor: corrosion rate calculated based on the stent volume loss after 5 weeks of immersion in bioreactor; Rabbit trachea implantation: corrosion rate calculated based on the stent volume loss after 4-weeks implantation in rabbit trachea. *denotes a significant difference between alloy groups ($p < 0.05$, $n = 3$). Corrosion rates calculated from bioreactor and rabbit trachea implantation are from the current study, the rest are derived from our previous published study[31].

stents remained after immersion of the stents for 5 weeks under the bioreactor flow environment in vitro. Degradation rate differences between the bioreactor used here, and the in vivo condition is due to stents experiencing cyclic stresses after implantation known to cause stress corrosion cracking (SCC)[48,49], eventually breaking down the stent structure. Further, a decrease in mechanical strength of the stent with magnesium matrix degradation causes stent weakening rendering fragmentation and collapse due to the stress generated from the tracheal movement further accelerating stent degradation. This is the first study utilizing the novel UHD alloy clearly warranting further research to understand the role of external static and cyclic mechanical loading stresses affecting the LZ61-KBMS stents degradation pattern. Currently there is no standard guidance indicating clinical time line for tracheal stent implants. Nevertheless, additional design optimization, microstructure control, and formation of corrosion-resistant biocompatible coatings could all be employed to further improve the durability of the implanted stents.

We failed to detect any remaining corrosion product in the μCT after complete degradation of the stent (i.e., at 8 and 12 weeks; Fig. 3F). It is well established that post-implantation of magnesium alloy scaffolds to an anatomical site leads to the formation of various hydroxides, oxides, and phosphates on the surfaces of the alloy which are in contact with the body fluids. The physicochemical nature of this deposited layer depends on various factors such as alloy composition, surface microstructure, implant site, volume and nature of the body fluid[50,51]. From the numerous published literatures it is well established that these in vivo deposited layers are generally thin and porous in nature and only give temporary corrosion protection to the underlying Mg surface. In this study, the LZ61 tracheal stents were exposed

to an open dynamic environment with moving mucus consistently cleaning the airway. The resulting dynamic condition around the implant surface influences mass transfer processes and hence, alters the local surface chemistry. This dynamic environment along with the complex mechanical movement of the trachea (e.g., swallowing, breathing, coughing, etc.) can further breakdown or can form cracks in the as-deposited magnesium-oxide-phosphate thin protective coatings. This can result in the rapid corrosion of the underlying bare LZ61 surface. Moreover, with the progression of the corrosion of the stent, the stent loses the structural and mechanical integrity and very likely forms many small fragments which further increases the bare alloy surface area and hence, the corrosion rate. This possibly explains the $53 \pm 12\%$ of the initial volume degradation of the stent after 4 weeks, and complete degradation observed within 8 weeks. The absence of corrosion products can be due to the prevailing dynamic conditions around the stent as well as the solubility of the corrosion products. It is expected that for the LZ61 alloy, the corrosion products under the tracheal environment are mostly consisting of chlorides, hydroxides, oxides, and phosphates of Mg and Li. The solubility of these salts in water is relatively much higher compared to similar salts of many transition metals and rare earths. Therefore, absence of the corrosion products in the trachea at 8 or 12-weeks' time point is not surprising.

The New Zealand white rabbit is an extensively used, proven model for pediatric subglottic injury and repair study[52–55]. The average rabbit subglottic diameter varies only minimally by weight, ranging from 5.4–5.8 mm, and correlates with the standard endotracheal tube (ETT) size for a 3–9-month infant (4.0 mm ETT, outer diameter 5.4 mm)[56,57]. Moreover, the rabbit model provides a mechanism to investigate laryngeal injury and repair over time, an essential paradigm for studying the pediatric airway in the context of airway growth[55].

After 12 weeks of implantation, distinct airway narrowing due to stenosis formation is clearly observed in the 316L SS stent group (Fig. 3C–E), typical for metallic non-degradable stents[47]. This is problematic because (1) it is impossible to remove or adjust the metallic stents without causing trauma; (2) the airway will be blocked again due to stenotic tissue ingrowth; (3) for pediatric patients, the metallic stent acts as a metal cage inhibiting the tracheal growth. The rabbit airway was fully restored to normal and continued to grow after the implanted LZ61-KBMS stent degraded. The LZ61-KBMS stents, therefore, have obvious advantages over the 316L SS stent due to the favorable degradation observed over time and more importantly, the non-induction of any noticeable stenosis on the surrounding tracheal tissue. The formation of hydrogen gas pockets around the LZ61-KBMS stents is a well-known phenomenon for magnesium alloy-based implants[44–46]. However, for tracheal stent application, $H_2$ gas pocket formation appears to be non-detrimental since the $H_2$ gas generated is easily removed during respiration.

The histological analysis clearly demonstrates the advantage of biodegradable LZ61-KBMS stents. In the first 4 weeks of implantation, the LZ61-KBMS stents exhibited comparable biocompatibility to the 316L SS stents based on the H&E and Alcian blue staining (Figs. 4 and 5). This was encouraging since 316L SS stents are widely used biomaterial implants with excellent biocompatibility in numerous device settings[58]. After complete LZ61-KBMS stent degradation, the tracheal tissue was normal compared to the healthy rabbit tracheal tissue as shown by both, endoscopic and histologic analyses. The tracheal mucosa was also fully restored at 8 weeks for the LZ61-KBMS stent. This is critical, since mucosa scarring can drive stenotic tissue formation. One of the biggest pitfalls raised by a previously reported study on degradable polymer stents is the possible migration of the

degrading stent remnants to the bronchi leading to progressing dyspnea[24]. In our study, all the rabbits survived well, and the lung histology showed no abnormalities caused by the degrading stent debris. We believe this is a major advantage of the LZ61-KBMS stents compared to the literature reports on biodegradable polymeric stents. Clusters of macrophages were observed in both metallic stent groups at week 4 (Fig. 6). With full degradation of LZ61-KBMS stent, no clusters of macrophages were observed indicating no obvious resolution of the inflammatory response to the stent at weeks 8 and 12. However, for the 316L SS stent group, clusters of macrophages were still observed at weeks 8 and 12 indicating the persistence of the inflammatory reaction to the stent and thus, a higher risk of fibrosis and trachea scarring evolution (Fig. 6).

These results are encouraging for the potential future application of this degradable metal stent, particularly among pediatric patients. Although biodegradable Mg alloy use in a rabbit tracheal model was shown in a recent publication[29], the pertinent studies were only conducted up until 8 weeks. Furthermore, the alloys contained rare earth elements and the alloy ductility appeared to be limited. Herein the LZ61-KBMS alloys are devoid of any rare earth elements and moreover, exhibit very favorable ductility characteristics rendering the system attractive from both biocompatibility and mechanical placement perspective for future tracheal applications. Despite the promising results, several questions remain to be answered. First, the full degradation process and ensuing degradation mechanisms of the LZ61-KBMS stent are still unknown. In this study, the LZ61-KBMS stent appeared to be eliminated between 4 and 8 week timeframe, which could involve a combination of corrosion and fracture. Second, the in vivo experiments were performed on healthy rabbits. Future studies should investigate the performance of the LZ61-KBMS stent in a tracheal stenosis model. Third, all the stents were delivered through an incision in the neck. A better non-invasive delivery system such as a balloon-based catheter with placement and expansion under imaging needs to be designed so that the LZ61-KBMS stent could be endoscopically implanted.

This study presents the preliminary data and results of in vitro and in vivo degradation, and tissue compatibility of Mg-Li-Zn alloy-based tracheal stent. Further studies are indeed required to systematically evaluate the clinical efficacy Mg-Li-Zn alloy-based tracheal stent. These include evaluating the mechanical properties of the stents vs. the clinical requirement for mechanical support of the trachea stents, a balloon delivery system to facilitate the non-invasive implantation, exploration of the stent in more clinically relevant animal models (e.g., SGS model), along with short and long term bio-toxicity studies combined with analysis of the corrosion product on the surrounding tissues and various organs.

In conclusion, we have demonstrated the feasibility of using the LZ61-KBMS, a novel ultra-high ductility (UHD) biodegradable magnesium alloy-based tracheal stent for airway placement and subsequent degradation to allow growth in a relevant animal model. This novel approach and the promising results presented herein may lead to a new treatment modality for airway obstruction. Clinicians may be able to apply tracheal stents without the concerns associated with the chronic presence of the stent or the morbidity associated with subsequent device removal. With subsequent study, the device design may be more refined, including optimized stent microstructure design as well as introduction of coating technologies to further improve the outcomes of this new technology.

## Methods

**Chemical composition analysis**. The actual elemental chemical composition of the fabricated Mg-Li-Zn-(Al) alloys were then verified by inductively coupled

plasma optical emission spectroscopy (ICP-OES, iCAP duo 6500 Thermo Fisher, Waltham, MA). Two standard solutions with known different concentrations of Mg, Li, Al, Zn, Mn, Fe, Cu, and Ni elements was prepared using certified single element standard solutions suitable for ICP (Sigma-Aldrich, St. Louis, MO). Deionized water was used as blank standard. Small pieces of the six alloys were cut from the extruded rod and dissolved in 1% nitric acid. The solutions were then diluted to analyze the concentration of elements mentioned above. The as determined chemical composition (wt. %) of the various elements are given in Supplementary Table 1.

**LZ61-KBMS tracheal stent fabrication**. LZ61-KBMS extruded rods were cut into mini-tubes, 4.2 mm in diameter, and 0.3 mm in wall thickness, and 60 mm in length using the wire-electrical discharge machining (EDM) approach with the help from the technical support team of the Advanced Manufacturing Techniques, Inc (Clifton Park, NY). AZ31 extruded rod was purchased from Goodfellow (Coraopolis, PA) and machined into the same dimension tube as well. Commercially available 316L SS mini-tube of identical dimension was purchased from Goodfellow (Coraopolis, PA). The mini tubes were fabricated using wire-EDM, and therefore, the tube surface was not seamlessly smooth but were devoid of any joints as shown in Fig. 1B (ii) and Fig. 1C (iv). The tubes were then later cut into stents and electrochemically polished to achieve a smooth surface before implantation (Fig. 1D (iii)) and described below.

The stent geometry was designed based on the mechanical properties of the LZ61-KBMS alloy with the support from our collaborator, Dr. Ke Yang's laboratory of the Institute of Metal Research, Chinese Academy of Sciences (Shenyang, China) as shown in Fig. 1C (ii). The stent design was developed in a 2D geometric pattern as a CAD file. The mini-tubes were later laser cut into stents based on this pattern with the technical support of Inotech Laser, Inc (San Jose, CA).

**Electrochemical polishing**. Electrochemical polishing of the stents was conducted using a self-designed, in-house set-up. This comprised of a DC power supply (Genesys™ Programmable AC/DC Power 2400 W, TDK-Lambda Americas, Neptune, NJ). A 250 ml glass beaker was used as the container of the electrolyte. A platinum wire was connected to the negative terminal of the DC power supply acting as the cathode. The LZ61-KBMS or AZ31 alloy stents were hooked onto a pure Mg wire while the 316L SS stents were hooked onto a 316L SS wire. The Mg wire or 316L SS wire was then connected to the positive terminal of the DC power supply acting as the anode. Both the anode and cathode were immersed in the electrolyte throughout the electrochemical polishing process. The composition of the different electrolytes used for the different stent materials are listed in Supplementary Table. 2.

The parameters of the electrochemical polishing process are listed in Supplementary Table 3. The current was fixed throughout the process and the total time of the polishing process was optimized experimentally. Due to the heat generated during the electrochemical chemical polishing process, dry ice was used to cool the electrolyte for the LZ61-KBMS stent and the AZ31 stent. Tap water was used to cool the electrolyte in case of the 316L SS stent. After electrochemical polishing, all the stents were cleaned using distilled water in an ultrasonic agitation bath for 15 min and were dried in air.

**Micro-computed tomography (μ-CT) imaging**. Micro-CT was primary used to analyze the volume of the stents to reflect the degradation profile and degradation rates of the corresponding stents, respectively. Stents were placed in a sample holder to prevent movement and scanned with continuous rotation μCT at 10.5 μm voxel size. The files were converted into the digital imaging and communications in medicine (DICOM) format and analyzed using the Mimics software (Materialise, Belgium). A threshold was set to isolate the implant from the surrounding tissues and the region growth function was used to identify the remaining structure of the stent which was then reconstructed into 3D images of all the stents. The volume of the electro-polished stents before implantation and on the harvested trachea was measured using a threshold of 220–650 Hounsfield units (HU) (Mimics Innovation Suite, Mimics Research 21, Materialise). In some cases, a region of interest was chosen to avoid any contribution from the surrounding cartilage. The Dynamic Region Grow tool was used to calculate the volume of the intact or fragmented stents. In order to verify the accuracy of the scanning as well as the analyses, at least ten different disk, rod, and pin-shaped Mg alloy samples with known length, diameter and volume (measured manually) were scanned and analyzed. The volume differences between the manually measured and μCT derived volumes were less than ±3%.

**Bioreactor setup**. ATA ElectroForce 3D CulturePro™ bioreactor setup (TA Instruments, New Castle, DE) was used to mimic the dynamic environment in vivo. Silicone tubes (Nalgene™ Pharma-Grade Platinum-Cured Silicone Tubing, Thermo Scientific) with 3/16 inches (4.76 mm) inner diameter, 1/16 inch (1.59 mm) wall thickness were used to mimic the tracheal tissue. The stents were mounted on a 5 mm × 2 cm balloon catheter (Mustang, Boston scientific), inserted into the silicon tube, and expanded using an inflation device (Acclarent®, Irvine, CA). The device continuously pushed saline into the balloon until a pressure of 12 atmospheres was

attained. The silicone tube with the stent inside was then hooked into the bioreactor chamber. In total, 3 LZ61-KBMS stents and 3 AZ31 stents were used, and each chamber only hosted one stent. Once the chamber was sealed with the stent insides, HBSS was pumped into the circulation system at the flow rate of 80 ml/min. All the stents were then scanned at 1, 3, and 5-week time periods using micro-CT. After completion of 5 weeks of immersion, the remaining stents were dried in ethanol and scanned under SEM (SEM; JSM6610LV, JEOL) and EDAX (EDAX Genesis, Mahwah, NJ).

**Procedures for implantation**. All the animal experiments were approved by the University of Pittsburgh's Institutional Animal Case and Use Committee (IACUC). 12-weeks old female New Zealand white rabbits (3.5–4.0 lbs.) were purchased from Covance Research (Denver, PA). All the rabbits were acclimated for 5 days before undergoing any invasive procedure, receiving the standard rabbit diet and water ad libitum. Before the surgery, the animals were anesthetized by IM (intramuscular) injection of ketamine/xylazine, 35 mg/kg and 5 mg/kg, respectively. The skin was prepared for surgery by clipping the hair and scrubbing the skin with betadine solution. For airway stenting, a vertical midline incision was made in the neck to expose the airway. The subglottis was entered via a midline cricoidotomy extending through the first and second tracheal rings, analogous to a laryngo-fissure procedure in a patient. The tracheal stent was then mounted on the balloon of the catheter (Mustang™, Boston Scientific). The catheter was then placed through the incision along with the stent. Upon reaching the target site, saline was injected into the balloon to expand the stent until 12 atmospheric pressure was reached within the balloon for 30 s. To further immobilize the stents and prevent stent migration, the stents were fixed with 4/0 Vicryl (polyglactin 910) interrupted sutures by suturing the stent to the tracheal wall. At the end of the procedure, the skin and sub-cutaneous layer together were closed with 4/0 Vicryl (polyglactin 910) interrupted sutures allowing air passage to prevent subcutaneous emphysema. Immediately after completion of the surgery, X-ray images were captured for all the animals to confirm the success of the surgical procedure.

Groups of 5 animals for LZ61-KBMS stents and 316L SS stents were used for each time point of 4, 8, and 12 weeks, respectively. The animals were sacrificed at the end of each time point by injecting with euthanasia drug of sodium pentobarbitol at the dosage of 50 mg/kg administered via intracardiac (IC) injection. Stented airway section, lung, liver, and kidney organ tissues were collected and fixed in 10% neutral buffered formalin for 3 days and then preserved in 70% ethanol.

**Endoscope**. A Parsons laryngoscope, Hopkins II 0° telescope (4 mm × 18 cm) and endoscopy tower (1288 HH definition camera and X8000 light source with fiber optic light cable, Stryker) were used to capture the stented airway of each rabbit immediately after the implantation and following 4, 8, and 12 weeks of implantation depending on the end time point.

**Optical coherence tomography (OCT)**. The fixe stented airway was scanned by OCT (Ilumien™ system, St Jude Medical) to examine the tracheal wall. The fixed trachea was immersed in PBS solution, and an image catheter (Dragonfly™ Imaging Catheter, St Jude Medical) was inserted from the distal end of the trachea all the way to the proximal end of the trachea. The catheter was then pulled back to scan the inner wall of the stented trachea which were controlled by the program.

**Histology analysis**. Fixed trachea and organs were dehydrated, infiltrated, and embedded in paraffin. The paraffin blocks with the tissues inside were then sectioned with a rotary microtome (Leica RM 2255, Leica Biosystems, Buffalo Grove, IL). These tissue sections were transferred to histology slides for histological staining. H&E staining was performed to examine the tissue morphology of the trachea and organs. Alcian blue staining was performed to examine the mucus and goblet cells within the mucosa. The CD68 (Ventana Medical Systems, Inc. Ventana/Roche Diagnostic, USA) marker was used to identify the location of macrophages in the airway tissue.

**Statistics and reproducibility**. The obtained results were expressed as the mean ± standard deviation. One-way ANOVA was conducted to determine the differences between the different groups of samples using the Bonferroni Procedure as post hoc test. Statistical significance was defined as $p < 0.05$. Statistical analysis was performed utilizing the IBM SPSS Statistics 23 package for Windows.

## Data availability

The manuscript contains datasets for the Figs. 2B; D; 3E; 5D and 7. Datasets for these figures are available at Figshare[59]. There are no restrictions on data availability. All remaining data would be available from the corresponding author (Prashant N. Kumta) upon reasonable request.

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

## Acknowledgements

The authors gratefully acknowledge the support of the NSF funded Engineering Research Center-Revolutionizing Metallic Biomaterials (RMB) through grant-EEC-0812348. PNK would also like to acknowledge the financial support of the National Science Foundation (CBET-0933153), the Edward R. Weidlein Chair Professorship Funds and the Center for Complex Engineered Multifunctional Materials (CCEMM) at the Swanson School of Engineering in the University of Pittsburgh for use of the equipment and instrumentation needed for execution of the work described herein.

## Author contributions

P.N.K. and J.W. conceived the project and the experiments. P.N.K., J.W., W.R.W., and D.C. conceived the in vivo experiments. J.W. conducted all of the LZ61-KBMS stent materials development and characterization experiments. J.W., A.R., Y.C., and B.L. conducted the in vitro and bioreactor experiments. D.C., L.M., J.W., A.R., and M.A. performed the in vivo experiments. J.W., B.L., and H.T.B. helped with the histology analysis. C.T. assisted with the OCT characterization. Stent design were provided by K.Y. and F.Z. J.W., A.R., L.M., B.L., and P.N.K. wrote the initial manuscript. The project was conceived and supervised by P.N.K.

## Competing interests

The authors declare no competing interests.
