## [Peer Review File · Communications Biology]

Reviewers' comments:

The authors fabricated LZ61 prototype airway stents and did some in vitro and in vivo tests to examine the clinical potential for pediatric laryngotracheal stenosis (LTS) treatments. Though the results were quite preliminary, this work could still provide some safety and biocompatibility information of the LZ61 airway stent. Here are the comments:

1. Actually, I do not think it is necessary to put the in vitro bioreactor degradation part in this manuscript. As is known to all that the static and dynamic degradation rate of Mg alloy were usually inconsistent. The authors here did not provide more insights to the underlying mechanism. If the aim was to better mimic the in vivo degradation situation, then the composition and the moving speed of the mucus etc. should be measured. Otherwise, it was only a coincidence that the dynamic degradation rate of the LZ61 was close to that of the in vivo case. Whether the bioreactor setup could be applied to other Mg alloys or other animal models was highly questionable.
2. The element composition of the LZ61 and AZ31 alloy should be given including the impurities e.g. Fe/Ni/Cu etc., which had a significant influence on the degradation rate of Mg alloy even at a very low concentration.
3. About making the mini tube, it seems the tubes were not seamless from the description. Please describe it more clearly.
4. About the in vivo degradation, it was a bit strange that the LZ61 airway stent was fully degraded after 8 weeks considering it was half remained at 4 weeks. Usually a compound of Ca/P/Mg/O was formed in situ when Mg alloy degraded. The degradation product usually took a long time to be fully metabolized. Could it be possible that the threshold set was inappropriate when conducting the micro CT segmentation? In Fig. 6 A (b) and (c), it seemed there were degradation products remained in the trachea.
5. Resin embedding was strongly recommended for preparing the slices not only for histological analysis but for degradation tests. It could provide direct evidence whether the degradation products were fully metabolized after 8 weeks. Besides, based on the SEM/EDS analysis results of the resin embedded slice containing Mg alloy implant, the segmentation threshold between implant/tissue, and degradation product/Mg alloy could be verified.
6. As the volume loss according to the micro CT results was used to quantify the in vitro and in vivo degradation rate, more details about how to determine the segmentation threshold should be given and verified.
7. As was realized by the authors, the current work could only provide limited efficacy information. There were many gaps: the known clinical requirement for mechanical support of the trachea by the stent, the attenuation curve of the stent mechanical properties, the animal model, etc.
8. It was suggested that OCT was conducted before and immediately after stent implantation to see the injury to the trachea (deduced from the trachea diameter change), which was critical to the subsequent host response.

The authors fabricated LZ61 prototype airway stents and did some *in vitro* and *in vivo* tests to examine the clinical potential for pediatric laryngotracheal stenosis (LTS) treatments. Though the results were quite preliminary, this work could still provide some safety and biocompatibility information of the LZ61 airway stent. Here are the comments:

1. Actually, I do not think it is necessary to put the *in vitro* bioreactor degradation part in this manuscript. As is known to all that the static and dynamic degradation rate of Mg alloy were usually inconsistent. The authors here did not provide more insights to the underlying mechanism. If the aim was to better mimic the *in vivo* degradation situation, then the composition and the moving speed of the mucus etc. should be measured. Otherwise, it was only a coincidence that the dynamic degradation rate of the LZ61 was close to that of the *in vivo* case. Whether the bioreactor setup could be applied to other Mg alloys or other animal models was highly questionable.

Response: The authors agree with the reviewer’s opinion that published *in vitro* data are, in most cases, inconsistent with the *in-vivo* results. However, we still believe that it is important to report the *in vitro* data to provide systematic visual and scientific evidence of the degradation process occurring at various time points. The *in vitro* setup in the bioreactor enables us to systematically follow the degradation of the stent on a weekly basis and gave valuable information about the relative degradation rates of the LZ61 and AZ31 alloy stents. It will of course be ideal if the same kind of systematic degradation study can be performed *in-vivo*. However, this will lead to a considerable increase in the numbers of animals used and will add to the exorbitant cost associated with it. We also understand that degradation results obtained from the bioreactor may not be comparable to other bioreactor based degradation studies or other *in-vivo* studies. However, publishing these *in-vitro* test results will give the scientific community an opportunity to compare these results with their outcomes and therefore, will provide a transparent and clear understanding of the underlying fundamental science and we believe will be a solid platform for future experiments and technologies. Moreover, various studies have compared and correlated the *in-vitro* and *in-vivo* degradation studies in bone and subcutaneous models but there is hardly any literature reported on the tracheal model [Acta Biomaterialia 13 (2015) 16–31] warranting publication of our results.

2. The element composition of the LZ61 and AZ31 alloy should be given including the impurities e.g. Fe/Ni/Cu etc., which had a significant influence on the degradation rate of Mg alloy even at a very low concentration.

Response: The chemical composition of the LZ61 alloys is given in the **Supplementary Information (Supplementary Table. 1)**. Since, the AZ31 alloy was procured from the commercial supplier (Goodfellow, Coraopolis, Pennsylvania, USA) no attempts were made to determine the chemical compositions using inductively coupled plasma optical emission spectroscopy (ICP-OES).

Table 1 The chemical composition of the LZ61

Chemical composition (wt. %)	Mg-6Li-1Zn
Li	6.11±0.13
Al	0.04±0.06
Zn	0.92±0.08
Mn	0.00835±0.00013
Fe	0.00045±0.0004
Cu	0.00040±0.00001
Ni	0.00018±0.000003

3. About making the mini tube, it seems the tubes were not seamless from the description. Please describe it more clearly.

Response: The mini tubes were fabricated using wire-electrical discharge machining (EDM) approach, and therefore, the tube surface was not seamlessly smooth but were devoid of any joints as shown in **Fig. 1B (ii)** and **Fig. 1C (iv)**. The tubes were then later cut into stents and electrochemically polished to achieve a smooth surface before implantation (**Fig.1D (iii)**). We have added this information in the **Supplementary Information** section.

4. About the in vivo degradation, it was a bit strange that the LZ61 airway stent was fully degraded after 8 weeks considering it was half remained at 4 weeks. Usually a compound of Ca/P/Mg/O was formed in situ when Mg alloy degraded. The degradation product usually took a long time to be fully metabolized. Could it be possible that the threshold set was inappropriate when conducting the micro CT segmentation? In Fig. 6 A (b) and (c), it seemed there were degradation products remained in the trachea.

Response: We agree with the reviewer that post-implantation of magnesium alloy scaffolds lead to the formation of various hydroxides, oxides and phosphates on the surfaces of the alloy which are in contact with the body fluids. The physicochemical nature of this deposited layer depends on various factors such as alloy composition, surface microstructure, implant site, volume and nature of the body fluid. From the numerous published literatures it is well established that these *in-vivo* deposited layers are generally thin and porous in nature and only give temporary corrosion protection to the underlying Mg surface. In this study, the LZ61 tracheal stents were exposed to an open dynamic environment with moving mucus consistently cleaning the airway. The resulting dynamic condition around the implant surface influences mass transfer processes and hence, alters the local surface chemistry. This dynamic environment along with the complex mechanical movement of the trachea (e.g. swallowing, breathing, coughing, etc.) can further breakdown or can form cracks in the as deposited magnesium-oxide-phosphate thin protective coatings. This can result in the rapid corrosion of the underlying bare LZ61 surface. Moreover, with the progression of the corrosion of the stent, the stent loses the structural and mechanical integrity and very likely forms many small fragments which further increases the bare alloy surface area and hence, the corrosion rate. This possibly explains the $53\pm 12\%$ of the initial volume degradation of the stent observed after 4 weeks, and complete degradation observed within 8 weeks.

We failed to detect any remaining corrosion product in the μ CT after complete degradation of the stent (i.e. at 8 and 12 weeks). A threshold of 220-650 Hounsfield units (HU) was used in the analyses to measure the stent volume before and post implantation. The absence of corrosion product can be due to the prevailing dynamic conditions around the stent as discussed earlier as well as the solubility of the corrosion products. It is expected that for the LZ61 alloy, the corrosion products under the tracheal environment are mostly consisting of chlorides, hydroxides, oxides and phosphates of Mg and Li. The solubility of these salts in water are relatively much higher compared to similar salts of many transition metals and rare earths. Therefore, absence of the corrosion products in the trachea at 8 or 12 weeks' time point is not surprising.

A paragraph has been added in the **Discussion** Section of the modified draft (**page 11, highlighted**) to address this issue raised by the reviewer. The details of the μ CT parameters are also added in the **Supporting Information**.

5. Resin embedding was strongly recommended for preparing the slices not only for histological analysis but for degradation tests. It could provide direct evidence whether the degradation products were fully metabolized after 8 weeks. Besides, based on the SEM/EDS analysis results of the resin embedded slice containing Mg alloy implant, the segmentation threshold between implant/tissue, and degradation product/Mg alloy could be verified.

Response: We agree with the reviewer that in some cases resin embedding samples offers unique advantages, especially histological processing of hard tissue such as bone. However, a large mechanical mismatch between the embedded tissue and the embedding material presented some unique challenges during histological sectioning, processing and staining. The local temperature rise due to resin polymerization process can also affect the thin soft tissues layers in the trachea. Bases on these observations, we preferred to use paraffin embedding over resin embedding in this study. Moreover, it is well known that performing immunohistochemistry on resin embedded samples are much more difficult than the paraffin embedded sample. Since the major objective of this work is to understand the safety and toxicity of the implanted tracheal stent, we have not performed a systematic *in-vivo* study on the stent degradation rate and on the dissolution-metabolization of the degradation products. We are thankful to the reviewer for these useful suggestions and our future study will correspondingly focus on these important issues.

6. As the volume loss according to the micro CT results was used to quantify the in vitro and in vivo degradation rate, more details about how to determine the segmentation threshold should be given and verified.

Response: The details of the μ CT parameters are added in the **Supporting Information** section. The volume of the electro-polished stents before implantation and on the harvested trachea were measured using a threshold of 220-650 Hounsfield units (HU) (Mimics Innovation Suite, Mimics Research 21, Materialise). In some cases, a region of interest was chosen to avoid any contribution from the surrounding cartilage. The Dynamic Region Grow tool was used to calculate the volume of the intact or fragmented stents. In order to verify the accuracy of the scanning as well as the analyses, at least ten different disk, rod and pin shaped Mg alloy samples with known length, diameter and volume (measured manually) were scanned and analyzed. The volume differences between the manually measured and μ CT derived volumes were less than $\pm 3\%$.

7. As was realized by the authors, the current work could only provide limited efficacy information. There were many gaps: the known clinical requirement for mechanical support of the trachea by the stent, the attenuation curve of the stent mechanical properties, the animal model, etc.

Response: The authors agree that the focus of the current study conducted was primarily to validate the feasibility of the novel Mg-Li-Zn alloy-based stent for tracheal stent study. This study presented the preliminary data and results of *in vitro* and *in vivo* degradation, and tissue compatibility of the Mg-Li-Zn alloy based tracheal stent. Further studies are indeed required to systematically evaluate the clinical efficacy of the Mg-Li-Zn alloy based tracheal stent. These include evaluating the mechanical properties of the stents vs. the clinical requirement for mechanical support of the trachea stents, a balloon delivery system to facilitate the non-invasive implantation, exploration of the stent in more clinically relevant animal models (e.g. subglottic stenosis model), along with short and long term bio-toxicity studies combined with analysis of the corrosion product on the surrounding tissues and various organs.

A paragraph has been added in the **Discussion** Section of the modified draft (**page 13, highlighted**) to address this issue raised by the reviewer.

8. It was suggested that OCT was conducted before and immediately after stent implantation to see the injury to the trachea (deduced from the trachea diameter change), which was critical to the subsequent host response.

Response: Thank you for suggesting this. Unfortunately, the surgery room was not equipped with an OCT machine to record the trachea diameter immediately after stent implantation. We therefore have no record of the post-surgery trachea diameter at 0 days. We only performed X-ray and endoscopy to assure proper placement of the stents. Nevertheless, the OCT analysis conducted following the surgical implantation provided valuable information attesting to the degradation of the implanted stent useful in crafting our final conclusions.

REVIEWERS' COMMENTS:

Reviewer #2 (Remarks to the Author):

The manuscript presents preliminary results about in vitro and in vivo degradation of tracheal stents produced by a novel Mg-Li-Zn alloy.

The results are original and the frame of information supplied by the paper will be of support for further studies.

The suggestions and comments raised by the reviewers have been substantially accepted and implemented in the new version.

I could just add a simple suggestion, based on text given at page 10, when discussing about the difference in corrosion rates of stents compared to previous studies. Three sources for discrepancy are suggested. I am quite doubtful about the last one concerning absence of heat treatment following wire-EDM and laser cutting of the stents. It has been demonstrated that the microstructural alteration due to above processing is very limited in thickness (few micrometers, if any) and this heat affected layer is usually fully removed by electrochemical polishing. Furthermore, any residual volume is soon lost by material degradation during the first stages of corrosion, so that possible effects due to the heat affected layer can be reasonably considered as fully negligible.

REVIEWERS' COMMENTS:

Reviewer #2 (Remarks to the Author):

The manuscript presents preliminary results about in vitro and in vivo degradation of tracheal stents produced by a novel Mg-Li-Zn alloy.

The results are original and the frame of information supplied by the paper will be of support for further studies.

The suggestions and comments raised by the reviewers have been substantially accepted and implemented in the new version.

I could just add a simple suggestion, based on text given at page 10, when discussing about the difference in corrosion rates of stents compared to previous studies. Three sources for discrepancy are suggested. I am quite doubtful about the last one concerning absence of heat treatment following wire-EDM and laser cutting of the stents. It has been demonstrated that the microstructural alteration due to above processing is very limited in thickness (few micrometers, if any) and this heat affected layer is usually fully removed by electrochemical polishing. Furthermore, any residual volume is soon lost by material degradation during the first stages of corrosion, so that possible effects due to the heat affected layer can be reasonably considered as fully negligible.

Response: The authors agree with the reviewer's opinion in most part. Heat treatment of metals is normally conducted to alter the bulk and surface microstructure to change the response of the material to the surrounding environmental influence. Typically, heat treatment results in changes in the bulk microstructure (grain size, shape and orientation) as well as distribution of secondary phases or precipitates within grains or at grain boundaries to result in changes in mechanical properties largely, strength and toughness. Surface related microstructures affected by heat treatment typically relate to surface composition and presence of oxidation layers on the surface initiated by heat treatment. Electropolishing could likely result in removal of surface related changes which tend to reside within a few micrometers. However, bulk related changes which are within the material will not be affected by electropolishing. Bulk microstructure can only be altered by secondary melting and solidification or extreme cold work and further refinement of the microstructure by heat treatment. We have accordingly modified the statement in the revised manuscript on page 7, last paragraph, line 7-9.